# Mechanical Control of Cell Migration by the Metastasis Suppressor Tetraspanin CD82/KAI1

**DOI:** 10.3390/cells10061545

**Published:** 2021-06-18

**Authors:** Laura Ordas, Luca Costa, Anthony Lozano, Christopher Chevillard, Alexia Calovoulos, Diala Kantar, Laurent Fernandez, Lucie Chauvin, Patrice Dosset, Christine Doucet, Lisa Heron-Milhavet, Elena Odintsova, Fedor Berditchevski, Pierre-Emmanuel Milhiet, Christine Bénistant

**Affiliations:** 1Centre de Biologie Structurale (CBS), CNRS, INSERM, University Montpellier, 34090 Montpellier, France; laura.ordas@inserm.fr (L.O.); costa@cbs.cnrs.fr (L.C.); anthonylozano0@gmail.com (A.L.); chris.chevillard@gmail.com (C.C.); alexia.calovoulos@outlook.fr (A.C.); laurentfernandez281288@gmail.com (L.F.); pat@cbs.cnrs.fr (P.D.); christine.doucet@cbs.cnrs.fr (C.D.); 2Institut de Génétique Moléculaire de Montpellier, University Montpellier, CNRS, 34293 Montpellier, France; 3Institut de Recherche en Cancérologie de Montpellier (IRCM), Inserm U1194—University Montpellier—Institut Régional du Cancer de Montpellier (ICM), 34298 Montpellier, France; diala.kantar@inserm.fr (D.K.); lisa.heron-milhavet@inserm.fr (L.H.-M.); 4European Institute of Chemistry and Biology (IECB), University of Bordeaux, 33607 Pessac, France; 5Centre de Recherche de Biologie Cellulaire de Montpellier (CRBM), CNRS UMR 5237, University Montpellier, 34293 Montpellier, France; lucie.chauvin@crbm.cnrs.fr; 6Institute of Cancer and Genomic Sciences, University of Birmingham, Edgbaston, Birmingham B15 2TT, UK; e.odintsova@bham.ac.uk (E.O.); F.BERDITCHEVSKI@bham.ac.uk (F.B.)

**Keywords:** cell migration, plasma membrane, tetraspanins, caveolins, focal adhesions, actin, mechanotransduction, YAP

## Abstract

The plasma membrane is a key actor of cell migration. For instance, its tension controls persistent cell migration and cell surface caveolae integrity. Then, caveolae constituents such as caveolin-1 can initiate a mechanotransduction loop that involves actin- and focal adhesion-dependent control of the mechanosensor YAP to finely tune cell migration. Tetraspanin CD82 (also named KAI-1) is an integral membrane protein and a metastasis suppressor. Its expression is lost in many cancers including breast cancer. It is a strong inhibitor of cell migration by a little-known mechanism. We demonstrated here that CD82 controls persistent 2D migration of EGF-induced single cells, stress fibers and focal adhesion sizes and dynamics. Mechanistically, we found that CD82 regulates membrane tension, cell surface caveolae abundance and YAP nuclear translocation in a caveolin-1-dependent manner. Altogether, our data show that CD82 controls 2D cell migration using membrane-driven mechanics involving caveolin and the YAP pathway.

## 1. Introduction

Cell migration is a necessary process during embryonic development and can be subverted in pathologies such as cancer metastasis. Cell migration is a complex and multi-factorial process allowing cells to migrate from a primary site to a secondary site, where they will proliferate and cause fatal lesions. Major steps in carcinoma metastasis are: (1) epithelial cells are released from cell–cell contacts by epithelial-to-mesenchymal transition (EMT); (2) cell migration across the basement membrane and underlying tissues; (3) intravasation into blood vessels and extravasation from blood vessels to the metastasis place; and (4) establishment of metastatic niches, mesenchymal to epithelial transition (MET), and dormancy or aberrant proliferation [1]. The existence of specialized program for metastasis formation is sustained by the discovery of metastasis-suppressor genes [2], which encode products that are specifically involved in cell migration and invasion for metastasis. Tetraspanin CD82, also named KAI1, falls into this category [3]. It is a member of the tetraspanin family (33 members) which are characterized by four transmembrane domains that separate two extracellular domains. Most tetraspanins are glycosylated on extracellular domains and palmitoylated at juxtamembrane intracellular cysteines. They are dynamic proteins diffusing within the plasma membrane and can assemble into dynamic tetraspanin-enriched microdomains in permanent exchange with the rest of the membrane [4].

CD82 is expressed in most cells and its expression is lost early during tumor progression in various tissues including prostate and breast cancers. Conversely, its presence indicates good prognostics for several cancers [5]. CD82 anti-metastatic effect has been linked to its ability to inhibit cell migration in Boyden chambers and in wound healing assays [6,7], to impact actin polymerization and Rho-GTPases signaling [8], EMT [9], clustering/signaling of integrins and receptors such as the epidermal growth factor receptor (EGFR), transforming growth factor β (TGFβ) and cMET [6,10,11], and to modify cell glycolipid composition [12]. In addition, CD82 regulates post-intravasation events such as cell elimination by DARC receptors in endothelial cells of blood vessels [13] and in the metastatic niche formation [14]. Its expression also inhibits EGF-induced 2D cell migration [6], further investigated in this work at the single-cell level. All these effects led us to speculate that CD82 may affect cell properties common to all these functions, especially those associated with the plasma membrane and we have considered membrane tension as a very good candidate.

Indeed, membrane tension has been shown to regulate many cell functions involving CD82, especially those linked to cell migration such as actin polymerization [15], cell polarization [16] or integrin clustering and positioning at the leading edge [17]. In vesicles, membrane tension arises from a combination of thermal fluctuation and lipid bending forces but, in cells, cytoskeleton organization also contributes to tension since the plasma membrane is attached to the underlying actin cortex [16]. In order to keep the membrane tension constant, cells use caveolae to buffer membrane tension changes during mechanical stresses [18]. They may also use microscale tetraspanin domains and migrasome formation as recently proposed [19]. Indeed, caveolae are small invaginations (60 nm diameter) of the plasma membrane that have the ability to flatten and disassemble upon an increase in membrane tension, releasing their constituents such as caveolin-1 and cavins [20]. Then, caveolin-1 associates to actin and integrins within focal adhesions to transmit mechanotransduction signals and promotes the translocation of the mechanosensor and transcription factor YAP into the nucleus [21]. YAP then controls F-actin polymerization [22], focal adhesion maturation and cell polarity in turn by modulating the balance of Rho GTPases RhoA/Rock [23], Rac1 [24] and Focal Adhesion Kinase (FAK) [25]. Importantly, one of the YAP transcriptional targets is caveolin-1, which in turn negatively controls YAP [26] and this creates a loop that tightly controls plasma membrane mechanics during cell migration.

Our working hypothesis in this study was that CD82 could control cell migration by impairing this loop. Our data clearly demonstrate that this is the case, with CD82 regulating membrane tension, focal adhesion, caveolae mechanotransduction and YAP signaling, and strongly support a central role of CD82 as an inhibitor of cell migration by controlling membrane mechanics during cell migration.

## 2. Materials and Methods

### 2.1. Materials

Hydrocortisone, cholera toxin, EGF, Concanavalin A, cytochalasin D, laminin, anti-tubulin and horse serum were from Sigma-Aldrich (St. Louis, MO, USA), France. Cell light Talin-GFP, phalloïdine-Alexa 555, insulin solution, Lipofectamine and Lipofectamine RNAiMax, DMEM, DMEM/F12, Prolong Diamond, Hoestch and Opitmem were from Thermo Fisher Scientific (Waltham, MA, USA). The mAb raised against CD82 has been described elsewhere [27]. Si RNA against human CD82 (KAI 1 siRNA (h) sc-35734), anti-caveolin-1 and anti-YAP antibodies were from Santa Cruz Biotechnology, SiRNA against human caveolin-1 (L-003467-00-0005) and human YAP-1 (L-012200-00-0005) from Dharmacon, SiRNA against human CD82 (GCTGGGTCAGCTTCTACAAdTdT, TTGTAGAAGCTGACCCAGCdCdG) and siRNA negative control (SR-CL00-05) from Eurogentec, fetal bovine serum from Gibco, MLCT-Bio-DC cantilevers from Bruker, paraformaldehyde from Electron Microscopy Sciences, poly-dimethylsiloxane (PDMS) from Neyko, France, Cytosoft plates from Advanced Biomatrix, protease inhibitors tablets from Roche Diagnosis, fluorodishes from World Precision Instrument, and anti-mouse star 635P was Abberrior. Sir Actin was from Spirochrome. The RNAeasy extraction kit was from Qiagen, primers for qPCR were designed and ordered at Eurofins Genomics or Eurogentec, Superscript III reverse transcriptase was from Invitrogen and the SYBR Green I Master mix used for qPCR was from Roche.

### 2.2. Cell Culture and Transfections

The establishment of HB2 and HB2-CD82 cells has already been described in [6]. Briefly, cells were extracted from human milk and immortalized using SV40 large T antigen. They were infected with an empty virus (HB2-Mock that we called HB2 for simplicity) or a virus coding for CD82 (HB2-CD82) and selected using puromycin. HB2 cells were cultured in a complete DMEM medium containing glutamine, 10% heat-inactivated fetal bovine serum (FBS), 1 mM pyruvate, 10 μg/mL hydrocortisone and 10 μg/mL insulin. MCF10a are derived from benign proliferative breast tissue and spontaneously immortalized without defined factors. They were cultured in DMEM/F12 containing glutamine, 5% horse serum, 10 µg/mL insulin, 20 ng/mL EGF, 0.5 µg/mL hydrocortisone, and 100 ng/mL cholera toxin and transfected with SiRNA by reverse transfection using RNAi max as described [28]. The absence of mycoplasma contamination was routinely checked using a MycoAlert Mycoplasma Detection kit from Lonza according to manufacturer instructions.

### 2.3. Live Cell Imaging

For migration experiments, cells were plated at low density in 24-well plates in 1% serum containing medium and cultured overnight. Then, the medium was changed and cells were stimulated with 100 ng/mL EGF and imaged on a Zeiss axio-observer equipped with an incubation chamber for temperature and CO_2_ controls (1 image every 15 min for 24 h). Single cells in the Appendix A were then tracked using MTrackJ (ImageJ) and migration data were analyzed using the DiPer program to obtain directionality ratio and average speeds cell by cell [29]. For Talin-GFP experiments, cells were incubated overnight according to manufacturer instructions in 1% serum, then stimulated with EGF and imaged using a spinning disk Nikon Ti Andor CSU-X1 or a Dragonfly Andor 1 in a CO_2_ and temperature-controlled incubation chamber. The 1 image/min during 20 min was acquired for HB2 cells and 1 image/min was acquired during 75 min for the MCF10a cells. Kymographs were extracted from the movies using the Fiji multi-kymograph program. For caveolin-GFP experiments, cells were transfected with the plasmid as described in [30] and imaged the day after using homemade objective-type TIRF setup equipped with a Plan Fluor 100×/1.45 NA objective (Zeiss, Le Peck, France Brattleboro, VT, USA). Wide-field imaging was used for YAP nucleo-cytoplasmic shuttling imaging using Zeiss AxioimagerZ1 or Zeiss AxioimagerZ2 upright microscopes equipped with Zeiss 40 or 63X Plan-Apochromat 1.4 oil objectives. Confocal images were taken using a Leica SP8 microscope equipped with a 63X NA1.4 oil objective using the LAS-AF software (Leica, Wetzlar, Germany).

### 2.4. Membrane Tether Measurements by AFM

Membrane tension was measured by tether extraction as described in [31]. In brief, plasma-cleaned MLCT cantilevers were coated using 2 mg/mL concanavalin A for 0.5–3 h, rinsed and mounted on the AFM head of a nanowizard 4 microscope equipped with the CellHesion module (JPK bioAFM, NanoBruker). Cells were plated on (40 µg/mL) laminin-coated 35 mm fluorodish at low density, grown overnight and placed in 1% serum medium before AFM experiments. Before each experiment, the cantilever spring constant (in N/m) was evaluated using the thermal noise method [32,33], whereas the inversed optical lever sensitivity (InvOls in nm/V) was evaluated through the acquisition of force curves onto a glass rigid surface. Cells were selected using phase-contrast microscopy and the cantilever was always placed at the maximal height above the nucleus. The following settings were used: 1 µm/s approach speed, 0.4 nN contact force, 2 s contact time, 3 µm/s retraction speed, 15 µm piezo amplitude, and 20 cycles per cell. Analyses were performed using the JPK software. Only curves where steps occurred at a distance over 2 µm of the surface were taken into consideration to avoid confusion with adhesion effects. In additional experiments described in S2, 3 cells were first measured as control. Then, the remaining cells in the dish were treated with cytochalasin D (200 nM final concentration) or by adding 1 volume of water (hypotonic shock) for 5 min and 3–4 additional treated cells were measured by a time not exceeding 1 h.

### 2.5. Ultrastructural Analysis by Transmission Electron Microscopy (TEM)

Cells grown on coverslip are fixed in 2.5% glutaraldehyde in phosphate buffer for 1 h RT and 24 h at 4 °C, post fixed with 1% osmium tetroxide, dehydrated and epon-embedded as described [30]. Infiltration is started by placing the coverslips in a 1:1 mixture of absolute ethanol and Epon 812 for 1 h and change for fresh resin 2 times 1 h. They were placed upside-down above an Eppendorf cap full of fresh resin and placed in a 65 °C oven. Then, Eppendorf caps were separated from coverslips by diving them in liquid nitrogen. Subsequently, Epon blocks were sectioned using an ultramicrotome (UC7 Leica) and 70 nm ultra-thin sections were mounted on 100-mesh collodion-coated copper grids, stained with uranyl acetate and lead citrate. The quality of the sections was checked by using low magnification micrographs (X1500). All the experiments were performed at the MEA platform of Montpellier University using a JEOL 1400 Plus electron microscope.

### 2.6. Fabrication of Polydimethylsiloxane (PDMS) Gels

We used either commercial gels Advanced Biomatrix or homemade gels prepared by mixing the dimethylsiloxane monomer with a cross-linking agent at a 1/60 ratio for soft (1–10 KPa) gels. After mixing and degassing under vacuum, the PDMS elastomers were spin-coated on glass coverslips or plates with a calculated volume to obtain a 50 µm thickness. Then, gels were cured for 2 h at 70 °C, UV sterilized, extensively rinsed with PBS, coated for 30 min with laminin (40 µg/mL) and rinsed again with PBS before being used for cell culture.

### 2.7. Immunofluorescence and Confocal Microscopy

Cells grown on glass coverslips or on PDMS gels, both coated with laminin, were fixed for 5–10 min at RT using a 3.7% fresh paraformaldehyde (PFA) solution. For actin studies, cells were permeabilized with PBS + 0.1% TRITON X100 for 2 min. After 2 PBS washes, they were incubated with (1/40) phalloidin-alexa555 for 20 min at RT and (1/1000) Hoestch for 2 min, washed several times in PBS and mounted on slides in Prolong diamond or Mowiol immersion solutions. For CD82 immunolabelling, Talin-GFP-expressing cells on glass coverslips were fixed for 5 min at RT with 3.7% PFA, washed in PBS and incubated in PBS supplemented with 3% goat serum for 10 min at RT. Then, cells were incubated with anti-CD82 (5 µg/mL in blocking solution) for 1 h at RT. After 3 washes with PBS, cells were incubated with anti-mouse star 635P (1/200 in blocking solution) for 1 h at RT, washed, fixed in 4% PFA for 5 min at RT, washed 2X with PBS then water and mounted in Vectashield. For YAP immunofluorescence studies, cells were permeabilized for 10 min at RT with PBS-0.3% TRITON X100. After PBS washes, cells were incubated with anti-YAP (1/200) in PBS-1% BSA for 6 h at 4 °C and then with anti-mouse alexa-647 (1/500) in PBS-1% BSA for 30 min at RT. Nuclei were labelled with Hoestch. Talin-GFP adhesion size quantifications were done on ImageJ as described in [23] using a 1–20 µm binning parameter. For actin fiber thickness quantifications, we first generated a plot-on line across cells on ImageJ and then quantified each peaks thickness on the plots.

### 2.8. Cell Lysates and Western Blots

Cells were lysed as described in [34]. Briefly, cells were washed in ice cold PBS then lysed in buffer containing 10 mM Tris pH 7.4, 150 mM NaCl, 0.5% TRITON X100, 60 mM octylglucoside, 10 µm NAF, 100 mM Vanadate and protease inhibitors. Lysates were cleared by centrifugation at 10,000 g, 10 min at 4 °C. Proteins in the lysates were separated by SDS-PAGE, transferred on nylon membranes and blot using indicated antibodies. Gel band quantifications were done on ImageJ.

### 2.9. RNA Extraction, Reverse Transcription and Real-Time RT-qPCR

RNAs were extracted from cells using RNAeasy plus kit (Qiagen) and cDNAs were prepared starting from 1 µg of RNAs using the Superscript III reverse transcriptase (Invitrogen) and following manufacturer directions. Real-time quantitative PCR was then performed on cDNAs using SYBR Green I Mastermix (Roche) according to manufacturer conditions on a Light Cycler 480 device (Roche). All primers used are listed as follows, 5′ to 3′:CYR61_For: ACCAAGAAATCCCCCGAACCCYR61_Rev: CGGGCAGTTGTAGTTGCATTCTGF_For: TTCCAAGACCTGTGGGATCTGF_Rev: GTGCAGCCAGAAAGCTCAREG_For: CGAAGGACCAATGAGAGCCCAREG_Rev: AGGCATTTCACTCACAGGGGBIRC2_For: GTCAGAACACCGGAGGCATTBIRC2_Rev: TGACATCATCATTGCGACCCACD82 For: ACTGGTTTCGTGGAAGGAAGCD82 _Rev: GCGCCCAGGATAAAGAAGATYAP_For: GCTACAGTGTCCCTCGAACCYAP_REV: ACTTGGCATCAGCTCCTCTC

### 2.10. Statistical Analyses

Anova analysis was used for simultaneously comparing means in several groups. Then, we used a 2-tailed unpaired t test or the parametric Mann–Whitney test for side by side comparisons between 2 groups. Numbers of * are indicated and represent the significance: 1 * *p* < 0.05, ** *p* < 0.01, *** *p* < 0.001, **** *p* < 0.0001.

## 3. Results

### 3.1. CD82 Regulates Persistent Cell Migration, Both Size and Dynamics of Focal Adhesions and Actin Polymerization

CD82 has been shown to inhibit cell migration in Boyden chambers or wound healing assays [6,14]. We further studied its effect on EGF-induced single-cell migration assay in 2D using the human breast-derived HB2 cells. These cells have lost the expression of CD82 in the course of cell immortalization and have been infected with an empty virus (named HB2) or with a virus expressing CD82 (named HB2-CD82) [35]. HB2-CD82 cells express a large amount of CD82 as compared to HB2 cells (Appendix A). This expression level, however, is in the range found in normal breast epithelium and represents over 5-fold the endogenous level found in MCF10a, the closest to normal human breast cell line (Appendix A).

We observed that HB2 cells that express a low level of CD82 (Appendix A) migrated preferentially in one direction, exploring a large area (Figure 1A and Appendix A). On the contrary, HB2-CD82 cells did not directionally migrate and have a tendency to turn around (Figure 1A and Appendix A). Accordingly, trajectories are longer in HB2 than in HB2-CD82 (Figure 1A) and both speed and directionality are impaired in HB2-CD82 cells (Figure 1B,C): the means +/−sem are 0.35+/−0.03 vs. 0.16+/−0.02 µm/min for speed and 0.30 +/−0.01 vs. 0.22+/−0.02 for directionality ratio in HB2 vs. HB2-CD82, respectively. Similar data were obtained in MCF10a (Appendix A) that constitutively express CD82 and turned around frequently (Appendix A), whereas MCF10a cells depleted of CD82 by siRNA had longer and straighter trajectories (Appendix A–E). As expected, the directionality ratio was increased from 0.21+/−0.13 to 0.35+/−0.17 in MCF10a siCtl vs. MCF10a siCD82, respectively (Appendix A). We have concluded that CD82 impacts persistent cell migration, an important result since this can drive metastasis formation in vivo [36].

As 2D cell migration relies on both focal adhesion and actin turnovers [37], we investigated the morphology of focal adhesions in HB2 and HB2-CD82 cells using the retroviral cell-light Talin-GFP (Figure 2A,B). We observed that HB2–CD82 cells have significantly more peripheral (yellow arrows in Figure 2A) and larger adhesion areas as compared to HB2 cells (the mean values ± sem of their surface are, respectively, 3.01 +/−0.78 and 3.78 +/−1.0 µm^2^) (Figure 2B). We also analyzed the dynamics of adhesion sites in Talin-GFP-expressing HB2 cells using spinning disk confocal microscopy and observed that focal adhesion in HB2 cells was more dynamic than in HB2-CD82 cells, retro-sliding away from the migration front as expected for a migratory cell (Appendix A). On the other hand, HB2-CD82 cells displayed very stable adhesions and did not migrate (Appendix A). Thus, CD82 might favor the formation of large mature and stable adhesion sites. Since focal adhesions are involved in building up actin stress fibers [38], we also performed actin labelling using the SiR-actin probe and observed that HB2-CD82 cells underwent a thickening of actin stress fibers (Figure 2C,D). In particular, the percentage of thin fibers (arbitrary chosen to be <1 µm diameter) over the total number of fibers was lower in HB2-CD82 as compared to HB2 cells (26.5+/−6.8 vs. 57.0+/−5.7; on the other hand, the number of thick fibers >1 µm was higher in HB2-Cd82 as compared to HB2 cells 73.4+/−5.7 vs. 42.9+/−5.7%; mean +/−sem).

Importantly, MCF10a silenced using two different SiRNAs (MCF10asiCD82 1 and 2) exhibited smaller (Appendix A) and more dynamic focal adhesions than MCF10a silenced with control SiRNA (MCF10aSiCtl) (Appendix A). In addition, the percentage of thick actin stress fibers (>1 µm apparent diameter) over the total number of fibers decreased in MCF10a silenced for CD82 as compared to the control (Appendix A). From these observations, we have concluded that CD82 regulates both focal adhesion dynamics and actin stress fibers architecture both in HB2 and MCF10a cells. In order to better understand how CD82 promoted these effects, we first assessed that CD82 in HB2 cells was largely expressed at the cell surface using flow cytometry on non-permeabilized cells (Appendix A). Interestingly, fluorescence microscopy experiments confirmed this and showed that CD82 mainly localized close to the Talin-enriched area and was often expressed in membrane projections that can arise from adhesions (appear as dotted structures using STED microscopy) (Figure 2E,F). A similar pattern in membrane projections was also observed in MCF10a cells (Figure 3E, lower panels). Taken together, these results clearly indicate that CD82 is expressed at the plasma membrane, where it can regulate cell migration, actin stress fiber sizes and focal adhesion dynamics.

### 3.2. CD82 Regulates Membrane Tension and Cell Surface Caveolae Density

Cell polarization and persistent cell migration are associated with an important remodeling of the plasma membrane that is associated with its deformability and change in its tension [16]. Recent data have also highlighted that membrane tension controls actin polymerization [15]. We thus investigated whether CD82 affects membrane tension.

To do so, we measured membrane tension in HB2 and HB2-CD82 cells by pulling membrane tubes using ConA-decorated AFM cantilevers as previously described [31] (Figure 3A,B). The force curves obtained during this process are then analyzed to extract the rupture force (squared in Figure 3A), which is the force required to break a membrane tether and relates to the membrane tension. The rupture force measured in HB2 cells (mean +/−sem) was 34.4 +/−3.6 pN and significantly different from 38.1 pN +/−3.9 measured in HB2-CD82 cells (Figure 3B). In MCF10a, we found that endogenous CD82 also regulates membrane tension. Indeed, the rupture force in MCF10a transfected with control SiRNA was 42.9+/−0.7 and decreased to 36.6+/−2.9 in MCF10a transfected with SiRNA targeting CD82 (Figure 3B).

We next compared this property of CD82 to that of a mild hypo-osmotic shock (see Section 2). We found a significant 1.2-fold increase in rupture force in HB2 cells incubated in hypoosmotic conditions as compared to non-treated cells, a magnitude of the decrease in good agreement with [17] (Appendix A). Surprisingly, rupture forces in HB2-CD82 cells were not significantly increased by the hypoosmotic conditions, suggesting that maximal tension could have been reached (Appendix A). Since membrane-to-cortex attachment control directed cell migration [16], we also probed the importance of actin cytoskeleton by measuring tether forces on cells treated with cytochalasin D, a well-known inhibitor of actin polymerization. We found a significant decrease in the tether forces under cytochalasin D treatment in HB2-CD82 cells (Appendix A), confirming the importance of the actin cortex in plasma membrane tension and in accordance with the changes in actin stress fibers morphology induced by CD82 described in Figure 2C,D. Since caveolae are membrane invaginations that behave as physiological membrane reservoirs and can buffer membrane tension induced by mechanical stress [18], we have then studied the effect of CD82 expression on the density of caveolae at the cell surface of HB2, HB2-CD82 and MCF10a cells. We first evaluated caveolin-1 expression levels in these cell lines by Western blotting and we found that caveolin-1 was expressed to a significant level in HB2, HB2-CD82 and in MCF10a cells (Appendix A). Next, we used transmission electron microscopy (TEM) on epon-embedded cells stained with uranyl acetate and lead citrate to observe caveolae at the plasma membrane. We found that plasma membrane of HB2 cells contains invaginations of both shape and size of caveolae (Figure 3C,D) and observed that the number of cell surface connected caveolae, measured by counting the number of open-neck caveolae per µm of membrane measured as described in [30], decreased 10-fold in HB2-CD82 cells as compared to HB2 cells (Figure 3C upper panels and Figure 3D). Mean+/−sem values were 2.0+/−0.4 and 0.2+/−0.1 open-neck caveolae/µm^2^ for HB2 and HB2-CD82, respectively. Similarly, we observed few vesicles of the size and shape of caveolae at the cell surface of MCF10 cells and their number significantly increased by ~ 3 fold at the cell surface of MCF10a transfected with siCD82 (1.1 +/−0.4 and 3.1+/−1 in MCF10a SiCtl and siCD82, respectively) (Figure 3C, lowers panels and Figure 3D).

We also measured the effect of a mild hypoosmotic shock on cell surface caveolae. Interestingly, we observed a reduction in cell surface caveolae in HB2 cells as described in [18] to reach the level observed in HB2-CD82 (Appendix A). In accordance with the observation of caveolae by TEM (Figure 3C), live-TIRF microscopy on caveolin-GFP cells indicated that HB2 cells present a lot of fast-moving GFP puncta close (deep to 200 nm) to the basal membrane surface that were decreased in intensity and speed upon a mild hypoosmotic shock (Appendix A), as described in [18]. By contrast, GFP puncta in HB2-CD82 were mainly less mobile (Appendix A), confirming that CD82 impacts caveolin-1-associated events. We also compared CD82 and caveolin-1-GFP localization in both non-permeabilized HB2 and MCF10a cells and observed that CD82 and caveolin-1-GFP were not colocalized (Figure 3E). We concluded that CD82 expression could impact the number of caveolae at the cell surface, consistent with its effect on membrane tension, but CD82 and caveolin-1 were not localized in the same compartment of the plasma membrane. A very recent work has shown that tetraspanins such as CD82 were involved in the formation of migrasomes, some cellular organelles that form as large vesicle-like structures, containing smaller vesicles, on retraction fibers of migrating cells [19]. Thanks to in vitro reconstitution experiments, it has been proposed that CD82 together with cholesterol could directly control membrane tension to generate these small vesicles. Interestingly, we also observed such structures by TEM in both HB2 and MCF10a cells (Figure 3F) and showed that the small vesicles in migrasomes show drastic changes in shape (Appendix A) and size as a function of CD82 expression, probably reflecting the changes in membrane tension induced by CD82, the migrasomes being likely less pro-migratory when CD82 was expressed and the vesicles deformed. Altogether, these data highlighted that CD82 regulates membrane tension and caveolae at the cell surface.

### 3.3. CD82 Regulates Caveolae Mechanosensing and YAP Nuclear Translocation and Activity

The transcription factor YAP is an important mechanotransductor in cells. In response to changes in cell rigidity or shape, it translocates into the nucleus to activate specific transcriptional programs [39]. Recent data have shown that YAP also regulates actin turnover by suppressing the formation of F-actin [22]. In addition, YAP regulates the mechanosensing of adhesion sites and, in turn, adhesions regulate YAP [23]. Moreover, YAP is also regulated by caveolae and, in turn, YAP regulates caveolin-1 expression [21]. Thus, YAP functions appear related to CD82 properties that we described above and we then investigated the effect of CD82 on YAP mechanotransduction in HB2, HB2-CD82 and MCF10a cells.

To do so, cells were grown overnight on commercial gel-coated coverslips or coverslips coated with homemade 50 µm thick silicone gels of 1–10 kPa rigidity and covered by laminin. We observed that on soft substrates, YAP was more present in the nucleus of HB2-CD82 cells as compared to HB2 cells, suggesting that HB2-CD82 cells have different mechanosensing properties than HB2 cells (Figure 4A,B).

This change in mechanosensitivity was also supported by the rounding up of the nucleus of HB2-CD82 cells (increase in the ellipticity ratio in Appendix A), a marker of mechanosensitivity [40] and a property that may impact YAP nucleo-cytoplasmic shuttling [41,42]. The percentages of YAP in the nucleus in cells grown on soft substrate (Figure 4B) were 28.0 +/−0.4 and 46.8+/−0.7 for HB2 and HB2-CD82 cells, respectively. However, no difference was observed in cells cultivated overnight on stiff glass coverslips (43.3+/−5.2 and 44.3+/−3.5 in HB2 and HB2-CD82, respectively).

Since caveolin-1 controls YAP nucleo-cytoplasmic shuttling and CD82 regulates caveolae shape and size, we next studied the contribution of caveolin-1 in CD82-induced YAP nuclear translocation on soft conditions by silencing caveolin-1 expression (Figure 4C–E). Achieving 75% silencing in HB2-CD82 cells (Figure 4C), the levels of YAP expression in the nucleus, displayed as the percentage of the total expression (mean+/− sem), significantly decreased from 66.0 +/− 4.1% in HB2-CD82 cells transfected with Ctl SiRNA to 39.9 +/− 5.8% in cells silenced for caveolin-1 (Figure 4E). A similar effect was also observed in MCF10a cultured on soft substrate (Figure 4F–H) where nuclear YAP in cells expressing the control SiRNA,- represents 46.1+/−7.6% of the total, this percentage decreasing to 32.1+/−6.8% in cells silenced for CD82 and to 26.4+/−3.0% in cells silenced for caveolin-1 (Figure 4H). In addition, we observed that CD82 endogenous level decreased in caveolin-1 MCF10a-silenced cells but this is unlikely to be due to YAP nuclear shuttling since CD82 was found not to be a target of YAP in our cell lines (see below). Since a similar effect was not observed in HB2 cells overexpressing CD82 (Figure 4C), it is likely that the stability of endogenous CD82 is impaired in caveolin-1 deficient MCF10a cells. Altogether, these data demonstrate that CD82 impacts cell mechanics and YAP nucleo-cytoplasmic shuttling mediated by caveolin-1.

YAP is a transcriptional co-regulator that can promote gene expression [39] and we next studied YAP transcriptional targets by RT-QPCR by measuring the expression of potential YAP targets in HB2 and MCF10a cells (Figure 5A, Appendix A). By downregulating YAP using SiRNA by 81 and 96%, respectively, in HB2 and MCF10a cells, we observed downregulations of several known YAP targets such as Cyr61, CTGF and AREG (Appendix A). However, we found that BIRC2 and CD82 are probably not targets of YAP in MCF10a (Appendix A). This hypothesis was also confirmed at the level of CD82 protein in YAP-depleted HB2 cells where CD82 expression was not significantly altered by YAP downregulation, whereas we did observe a net increase in YAP expression in caveolin-1-depleted HB2-CD82 cells (Figure 4C) as described [21]. Although YAP downregulation was strong, the amount of target genes was only decreased by about 50%. This is probably because YAP was highly expressed and that the remaining YAP is sufficient to exert some transcriptional activity. In addition, it is generally necessary to downregulate both YAP and TAZ to achieve a complete transcriptional inhibition [22]. We also analyzed the effect of CD82 expression on the validated YAP targets. We first observed that CD82 regulates YAP targets in both cell lines (Figure 5A and Appendix A). For example, AREG mRNA is increased in CD82-overexpressing HB2 cells (1.42+/−0.28, mean+/−sem, n = 3 experiments) and decreased in MCF10a cells knock down for CD82 by siRNA (Figure 5A and Appendix A). As another example, CYR61 mRNA levels are decreased down to 0.29%+/−0.01 (mean+/−sem, n = 3 experiments) by CD82 knockdown in MCF10a as compared to control (Figure 5A). Interestingly, knock down experiments in MCF10a give stronger effects when compared to overexpression experiment, probably because cells can better tolerate CD82 overexpression. However, we did not observe major differences in YAP target gene expression in stiff vs soft conditions. On one hand, this is not surprising since a ~95% reduction in YAP expression induces only a 50% target gene inhibition (Appendix A) while we expected only a 10% difference for a 25% difference in nuclear YAP between stiff vs soft conditions. On the other hand, this indicates that nuclear YAP activity in stiff conditions is still regulated by CD82 and YAP has probably not the same activity when CD82 is present.

Finally, we next studied whether YAP was required for the CD82 effects and we first studied its effect on MCF10a cell motility. We demonstrated that YAP-silenced cells explore a larger area in accordance with an increased in directionality (Appendix A–E), a phenotype similar to the one observed in CD82-depleted cells. In addition, depletion of YAP induces a reduction in Talin-GFP adhesion sizes induced by CD82 in MCF10a (Figure 5B,C), with mean values +/−SD 3.9+/−3.2 and 2.8+/−2.6 for MCF10a siCtl and siYAP, respectively, and HB2-CD82 (Figure 5D,E) mean values +/− SD 4.0 +/− 3.7 and 3.4 +/− 3.4 for HB2-CD82 siCtl and siYAP, respectively, indicating that YAP downregulation can revert the CD82 effect on focal adhesions. These results indicated that YAP is required for the effect of CD82 on adhesion sites.

## 4. Discussion

Cell mechanical properties regulate cell migration and can be finely tuned depending on the microenvironment. They can contribute to pathologies such as cancers when deregulated [43]. Our work reveals that expression of the metastasis suppressor tetraspanin CD82/KAI1, indicative of good prognosis in breast cancer, is a regulator of cell mechanics in breast epithelial cell lines. We propose that it regulates single-epithelial 2D cell migration through tuning of membrane tension which can induce caveolae disassembly and caveolin-1 release. Through the caveolin pathway, the effect of CD82 expression on cell mechanical properties leads to the activation of the mechanosensor YAP/TAZ signaling pathway and YAP translocation into the nucleus. Our model (Figure 6) proposes that CD82 expression may contribute to maintain epithelial cells into an immobile state. Conversely, early loss of CD82 expression during carcinoma formation may contribute to release this constraint inducing migration of epithelial cells and can explain part of the metastasis-suppressor functions of CD82.

As described above, we believe that the decrease in the cell surface density of caveolae induced by CD82 is the consequence of its ability to increase membrane tension. Even if we cannot exclude that CD82 could regulate membrane tension by directly affecting the caveolae system, it seems very unlikely since, even if both CD82 and caveolin are associated with membrane microdomains, no relationship between these two proteins within the plasma membrane has been clearly established. They are very rarely found to interact with each other (to our knowledge, only one paper describes their interaction [44]) and they are expressed in different membrane compartments. This interpretation is also supported by our results showing that CD82 is localized at the periphery of focal adhesions in some membrane projections where caveolin-1-GFP is not expressed. Interestingly, these projections are sometimes identified as migrasomes, some recently discovered cellular organelles that form as large vesicle-like structures [19], and we have clearly observed such structures in HB2 cells and showed that the shape of vesicles into the migrasomes are modified according to CD82 expression (EM picture in Figure 3F and Appendix A). This may be a consequence of CD82 function in regulating membrane tension (CD82 is involved in the migrasome formation and/or could impair the release of its constituents that contributes to cell migration). It is tempting to propose that CD82 could act on membrane tension through the tetraspanin web but, at the same time, it is the unique tetraspanin clearly described so far as a metastasis suppressor. Indeed, CD81 functions in breast cancers are not clear, its expression being correlated to either bad or good prognosis [45]. Concerning CD9, we and others have shown that it can control cell migration but also often proliferation such as in parietal epithelial cells [46], as opposed to CD82 that is not affecting proliferation. Another interesting candidate is TSPAN7 that has an anti-tumor effect in some cancers such as bladder cancer [47] but not in others such as lung cancer [48].

Among the components of the plasma membrane that could be involved in CD82 regulation of membrane tension, lipids are also good candidates, especially cholesterol and glycosphingolipids (GSL). Indeed, cholesterol largely influences membrane order within biological membranes (reviewed in [49]), was recently proposed as a regulator of membrane tension by FLIM microscopy using the lipid probe FliptR [50] and it is also a key component of tetraspanin assemblies containing CD82. A direct link between CD82 and cholesterol was demonstrated by CD82 labeling with a photoactivatable cholesterol in vivo [51] and cholesterol/CD82 interplay is also supported by a recent publication showing that both CD82 and cholesterol are necessary and sufficient for migrasome formation during cell migration, modifying membrane line tension [19]. We also question the implication of GSL that are known to play an important role in the organization of CD82-enriched microdomains [12] and can modulate cell motility in epidermoid carcinoma cells [52]. GSL could regulate membrane tension through the glycosynapse via their glycosylation motives [53]. EGFR activity is sensitive to the level of GM3, a GSL that is regulated by CD82 [53], and this may contribute to the CD82-induced regulation of cell migration under EGF through the regulation of EGFR activity and endocytosis, maybe through the modulation of its dynamics and signaling as described in [54]. GSL could also link indirectly CD82 to caveolae since we previously found that the GSL GM1 is directly involved in the control of cell surface caveolae in fibroblasts [30] and it has been recently reported that caveolae transport gangliosides [55]. To finish, we consider one candidate that is specific for CD82 so far and named Kitenin, also called planar cell polarity protein or Van Gogh-like (Vangl)-1 protein [56]. This protein is a core component of the non-canonical Wnt planar cell polarity pathway that controls epithelial polarity and cell migration and that can play important role in collective movement for tissue organization and in cancers [57]. This protein interacts with CD82 and is structurally related to it with four transmembrane domains but shorter extracellular loops and it would be interesting to test its contribution to the CD82 effects.

We also found that CD82 regulates caveolin-1-induced YAP nucleo-shuttling on soft substrates and YAP transcriptional activity on both soft and stiff substrates. YAP nuclear localization and activity are tightly regulated by mechanisms that are still not completely understood. For example, a very recent work showed that it is the dynamics of nuclear YAP entry and exit that control YAP activity and not only its presence/absence in the nucleus [58]. Another work has shown that cell contractibility promotes tyrosine phosphorylation on additional YAP tyrosine residues by Src kinases, within the nucleus, and that this regulates its nuclear exit and that YAP nuclear exit is the limiting factor of YAP nucleo-shuttling [42]. Clearly, it will be very interesting to study the effect of CD82 on YAP nucleo-shuttling dynamics. We also believe that it could be interesting to study YAP post-translational modifications using proteomics. Indeed, YAP in the nucleus is generally associated with cell transformation and tumors, but is here surprisingly associated with anti-migratory properties. YAP is known to be regulated by many post-translational modifications (70 sites described in Phosphosite plus^R^). Maybe CD82 favors a combination of these YAP post-translational modifications that drive YAP to induce an anti-migratory gene induction program that would be interesting to analyze using transcriptomic. However, YAP also has some functions independent from transcription and we observed a stronger effect of YAP downregulation on adhesion sizes that on its classical target genes mRNA regulation. Maybe YAP localization in the nucleus is also to deplete it from the cytoplasm and to regulate its cytoplasmic functions as we recently described for the junctional protein MAG1 [59]. Additionally, there exists up to eight different YAP1 isoforms due to splicing and some may have different functions [60]. Clearly, YAP functions and regulations are complex and more studies are needed to understand the full picture of CD82 function in breast epithelial cells.

As a conclusion, our results show that the anti-metastatic property of CD82 is probably a consequence of its impacts on membrane tension and cell mechanics, linked to caveolae mechanosensing and to the YAP/TAZ pathway.

## Figures and Tables

**Figure 1 cells-10-01545-f001:**
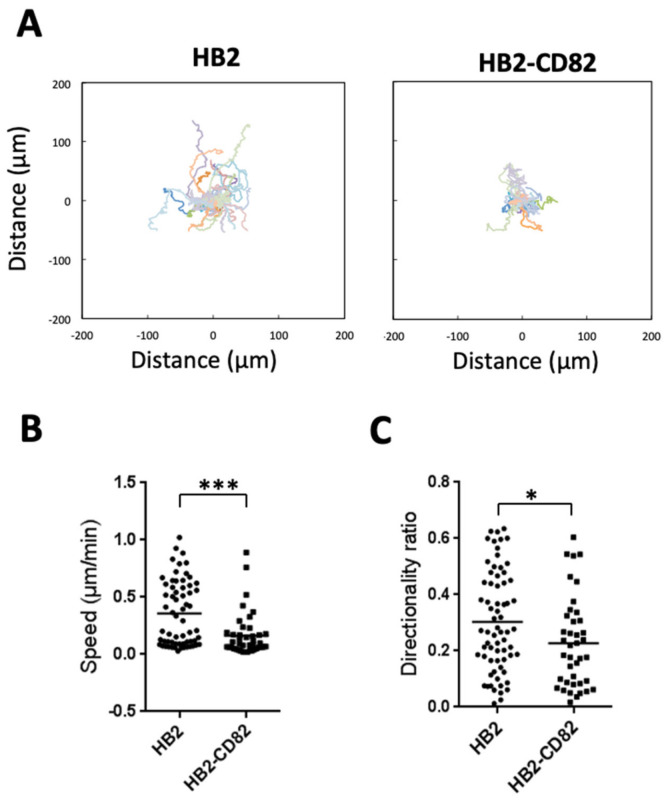
CD82 regulates EGF-induced single-cell persistent migration. (**A**) Trajectories of the 10 EGF-stimulated single HB2 or HB2-CD82 cells; (**B**) average speed of the cells; (**C**) directionality ratio of the cells determined as the shortest distance between the starting and arrival points divided by the real distance traveled by the cells (n = 68 and 40 cells, respectively, for HB2 and HB2-CD82, obtained from 10 independent experiments), * *p* < 0.05, *** *p* < 0.001.

**Figure 2 cells-10-01545-f002:**
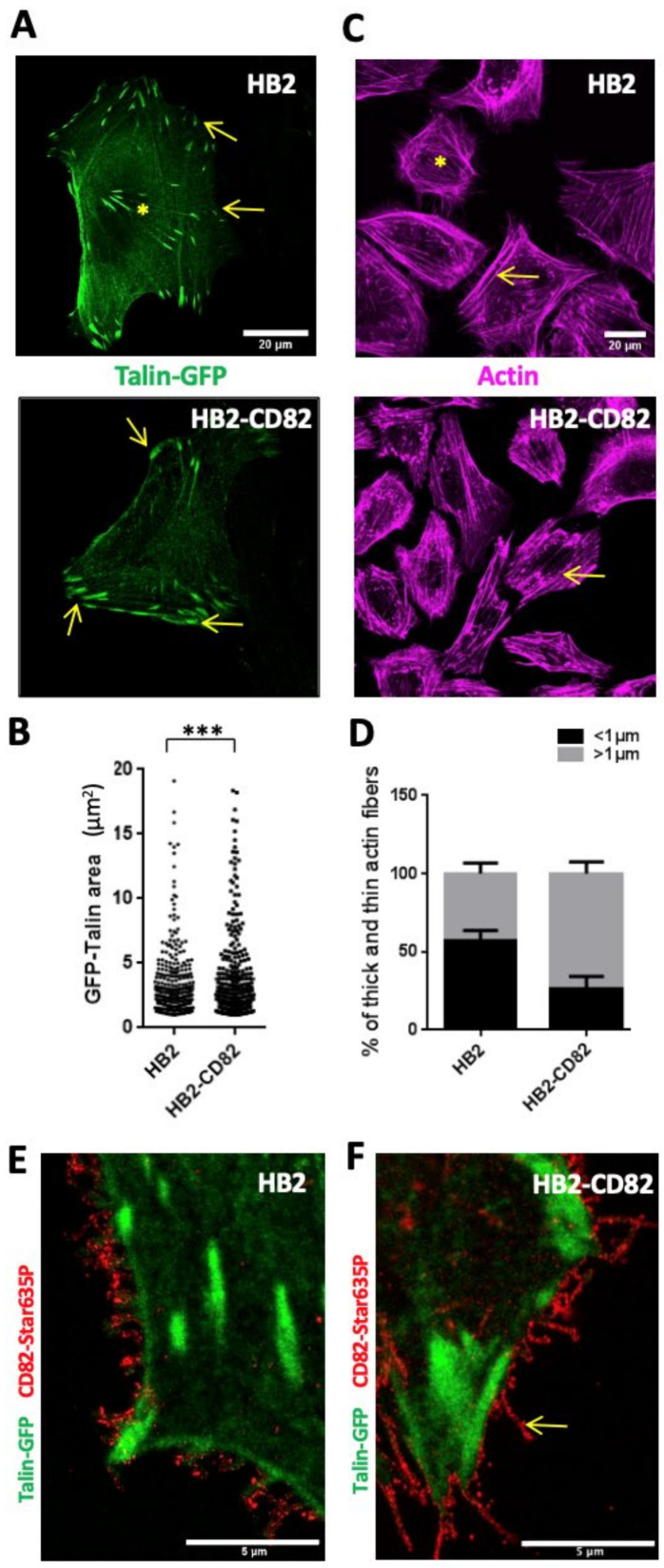
CD82 regulates both size and dynamics of focal adhesion and actin fiber sizes. (**A**) Talin-GFP labelling in HB2 and HB2-CD82 (left panel). Arrows indicate peripheral adhesions that are smaller in HB2 cells than in HB2-CD82 cells. The yellow asterisk indicates small central adhesions in HB2 cells. (**B**) Quantification of the size of the peripheral adhesions on 11 cells from 4 independent experiments, *** *p* < 0.001. (**C**) SiR actin labelling of HB2 cells and HB2-CD82 cells (right panel). Arrows point out thick fibers in HB2 and HB2-CD82 and asterisks small fibers in HB2 cells, respectively. (**D**) Quantification of fiber thickness using the Fiji plugin “line plot” (measurement of the width of the picks), shown as the percentage of the total number of fibers on 20 cells from 4 independent experiments (the fibers are arbitrary classified as thin or thick if their thickness is less than or greater than 1 µm, respectively), (**E**,**F**) Fluorescent images of CD82 (STED) and Talin-GFP (confocal) in HB2 and HB2-CD82 cells, respectively. Bar = 5 µm. The yellow arrow points out CD82 in a membrane extension in HB2-CD82 cells.

**Figure 3 cells-10-01545-f003:**
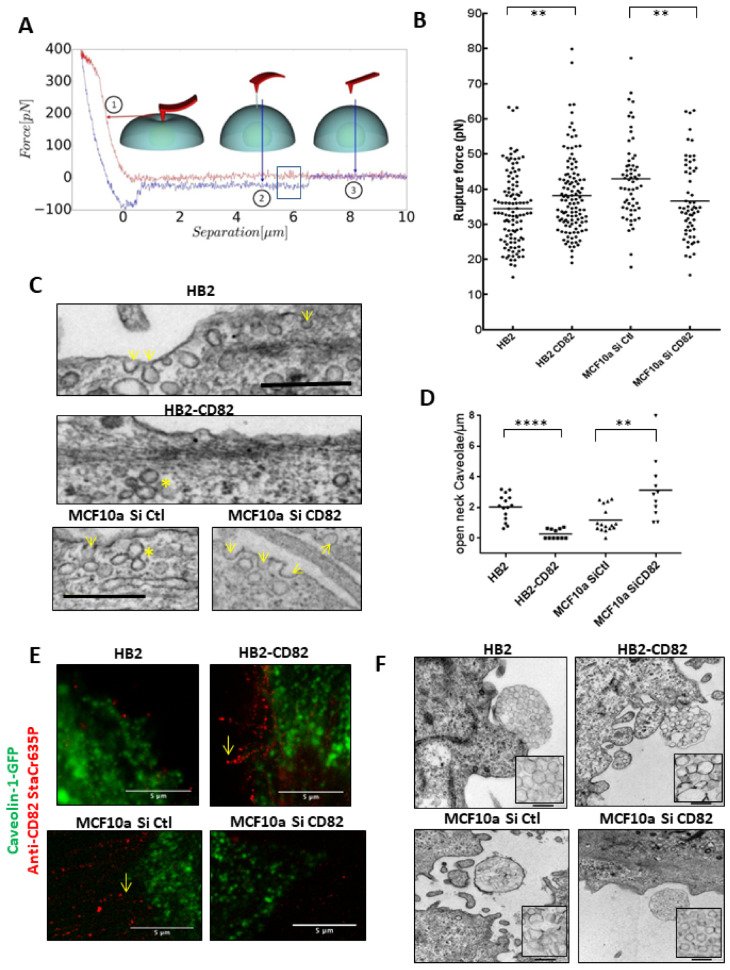
CD82 regulates membrane tension and the presence of cell surface caveolae in HB2 and MCF10a cells. (**A**) Schematic view of the membrane tether pulling experiment. The approach and retract curves are, respectively, in red and blue. Step 1 corresponds to the tip indenting the cell membrane, Step 2 to the pulling of the membrane tube during retraction and Step 3 to non-contact between the tip and the membrane. The blue rectangle highlights the rupture event between the tube and the tip where the rupture force is measured. (**B**) Rupture force of plasma membrane tubes as a read out of membrane tension in HB2 (110 measurements, 8 different probes, 21 cells, n = 8) and HB2-CD82 cells (119 measurements, 8 different probes, 19 cells, n = 8) and in MCF10a cells transfected with control siRNA (Si Ctl, 61 measurements, 2 different probes, 7 cells, n = 3) or with siRNA targeting CD82 (Si CD82, 60 measurements, 2 different probes, 8 cells, n = 3); ** *p* < 0.01. (**C**) TEM microscopy images of membranes of HB2 (upper), HB2-CD82 (middle), MCF10a cells transfected with siRNA control (lower left) or with siRNA targeting CD82 (lower right). Arrows indicate open-neck caveolae; * indicate rosettes; bar = 500 nm. (**D**) Quantification of the number of open-neck caveolae observed by EM and expressed per micrometer of membrane. **** *p* < 0.0001; (**E**) Fluorescent images of CD82 (STED) and Caveolin-GFP (confocal) in HB2 (upper, left), HB2-CD82 (upper right), MCF10a si Ctl (lower left) and MCF10a si CD82 (lower, right) cells. The yellow arrow points out CD82 in a membrane extension in HB2-CD82 cells and MCF10a si Ctl. (**F**) TEM microscopy images of migrasomes in HB2, HB2-CD82, MCF10a si Ctl and MCF10a siCD82 (bar = 500 nm); inserts enlighten vesicles inside migrasomes.

**Figure 4 cells-10-01545-f004:**
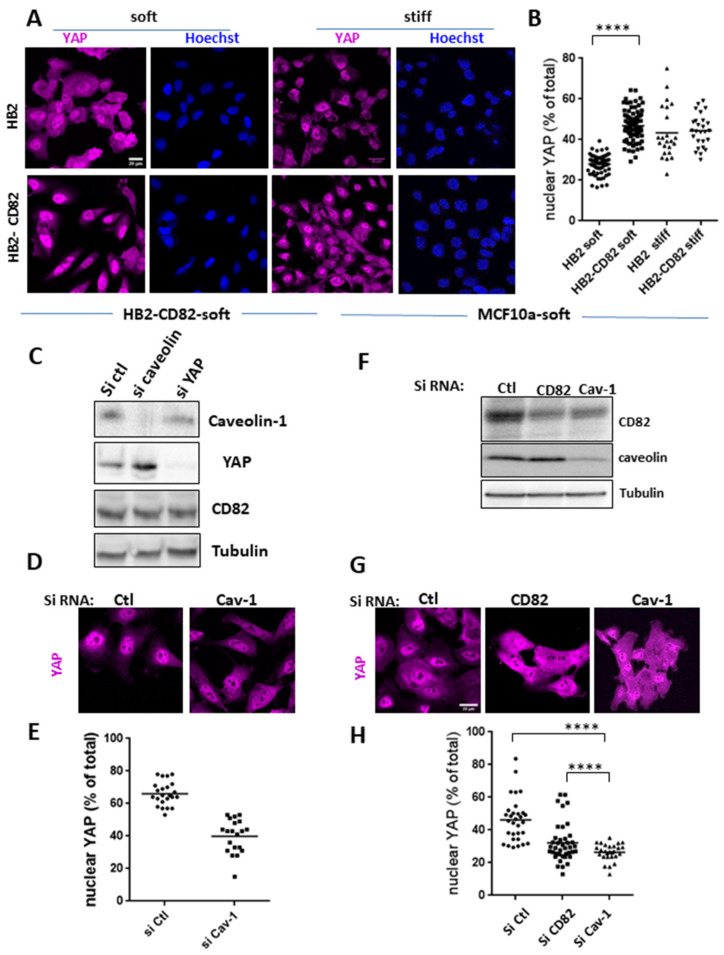
CD82 regulates caveolin-dependent YAP nuclear translocation. (**A**) Glass coverslips were coated with (soft, 1 kPa) or without (stiff, GPa) silicone gels then with laminin. HB2 and HB2-CD82 cells were cultivated overnight on these coverslips, then fixed and labelled with YAP antibodies or Hoestch to visualize nuclei (soft silicon: n = 73 and 84 cells in HB2 and HB2-CD82 cells, respectively, n = 10 experiments; stiff silicon: n = 23 and 26 cells, n = 4, respectively). Bar = 20 µm. (**B**) Nuclear YAP expressed as a percentage of the total amount; **** *p* < 0.0001 ns: non-significant. (**C**) Caveolin, YAP and CD82 expressions in HB2-CD82 cells transfected with siRNA control (siRNA Ctl), SiRNA targeted caveolin-1 (si Cav-1) or SiRNA targeted YAP (Si YAP) grown 24 h on soft silicon gels coated with laminin and lysed. Tubulin is shown as a loading control. (**D**) Same experimental conditions as in C but cells are fixed and labelled with anti-YAP. Bar = 20 µm. (**E**) Quantification of images shown in D as described in B using Hoestch labelling (not shown) to visualize nuclei (23 and 20 cells, n = 3 for HB2-CD82 Si Ctl and HB2-CD82 Si Cav-1): **** *p* < 0.0001. (**F**) CD82 and caveolin-1 expression in MCF10a si Ctl, siCD82 and Si Cav-1 cultured on soft substrate (MCF10a-Soft). Tubulin is shown as a loading control. (**G**) YAP localization in MCF10a transfected with siRNA Ctl, siRNA targeted CD82 or Caveolin-1 and cultured on soft substrate: bar = 20 µm. (**H**) Quantification of nuclear YAP expression in MCF10a Si Ctl (n = 31 cells), Si CD82 (n = 40) and Si Cav-1 (n = 27) from 3 separate experiments.

**Figure 5 cells-10-01545-f005:**
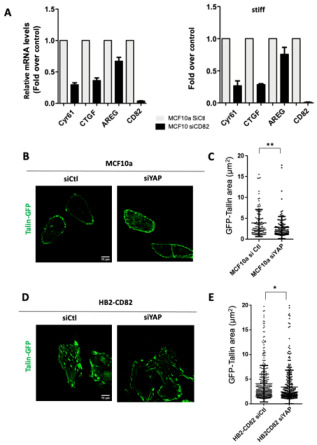
CD82 regulates and depends on YAP. (**A**) Representative RT QPCR analysis of YAP target genes in MCF10a transfected with Si Ctl or Si CD82 cultured on soft (left) and stiff (right) substrates (one experiment is shown here out of 3, done in triplicate). (**B**) Talin-GFP labelling in MCF10a cells transfected with Si Ctl or Si YAP **C**) Quantification of the size of the peripheral adhesion areas of 11 cells from 2 independent experiments; ** *p* < 0.01. (**D**) Talin-GFP labelling in HB2-CD82 cells transfected with Si Ctl or Si YAP (**E**) Quantification of the size of the peripheral adhesion areas of 13 and 14 cells, respectively; * *p* < 0.05.

**Figure 6 cells-10-01545-f006:**
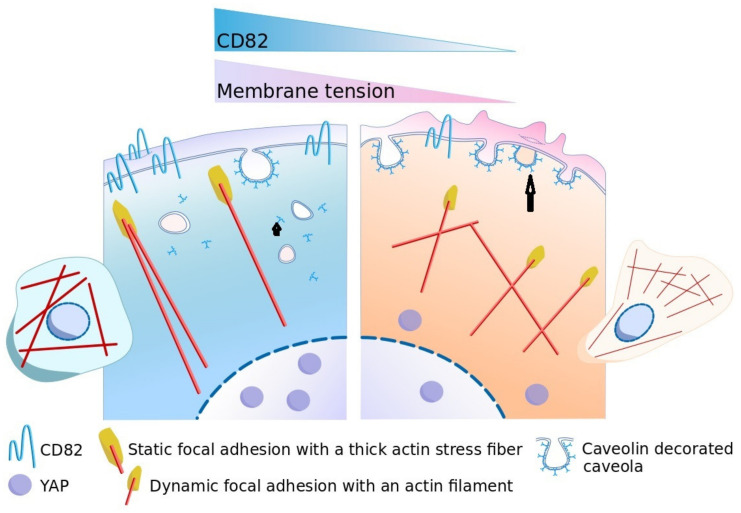
Model of CD82 action on 2D cell migration of epithelial cells. Left: In normal situations, CD82 is expressed and membrane tension is high; there are a few caveolae at the cell surface due to membrane tension buffering [18] and free caveolin-1 (black arrowhead). Free caveolin-1 can initiate a mechanotransduction cascade that drives YAP into the nucleus [21]. In turn, YAP controls caveolin-1 expression [26] and both YAP and caveolin-1 control focal adhesions [21,23] and actin polymerization [21,22] which also control membrane tension [16] and caveolae formation, looping a loop that keeps cells immobile. Right: When CD82 is poorly expressed, such as in breast carcinoma, the early loss of CD82 leads to a decrease in membrane tension, and an increase in cell surface caveolae (black arrow), allowing membrane tension fluctuations that, in synergy with actin polymerization fluctuations underneath the membrane [15] and with focal adhesion positioning at the leading edge [17], can form a stable lamellipodia required for persistent cell migration.

## Data Availability

All the data provided in the manuscript and Appendix A.

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
