# Peer review of "Mechanical Control of Cell Migration by the Metastasis Suppressor Tetraspanin CD82/KAI1"

_cells, 2021, doi:10.3390/cells10061545_

Round 1
Reviewer 1 Report
In this work, the authors studied the impacts of the tetraspanin CD82, which has some known anti-metastatic effect in tumors, on the surface membrane tension. Their starting hypothesis is that CD82 could impact surface membrane tension and cell migration by being involved in the known loop between caveolae, with a role in membrane buffering during changes of membrane tension, and the transcription factor YAP which is also a mechanosensor. In order to do so, they used two breast cell lines: HB2 cells overexpressing CD82 and MCF10a cells where they are knocking down CD82 by siRNA. Then they are measuring, cell migration (speed, directionality), size and dynamic of focal adhesion plans, membrane tension and are imaging by confocal microscopy, EM and TIRF, caveolae localization and distribution, alongside YAP localization and activity as a transcription factor. Based on their data, the authors proposed that CD82 controls 2D migration of EGF-induced cells, stress fibers and focal adhesion size and dynamics by impacting cell surface caveolae abundance and subsequently YAP nuclear translocation. Although the data proposed here are going in the right direction to support the claims made by the authors, they are often lacking more rigorous analyses (quantifications) alongside some other measurements/conditions in order to reinforce their conclusions. Although the manuscript is clear and well written, figure panels and supplementary figure panels are not totally following the order in which they are cited in the text, making the reading more difficult.
Major comments:
Introduction:
-Line 74: Are Caveloae the only process used by cells to buffer membrane tension? From this introduction it seems like this whereas it is certainly not. Could the authors correct this aspect?
- CD82 is not the only member of the tetraspanins to be involved in membrane tension and actin rearrangement. Could they mention in the introduction the roles of other members such as TSPAN7 for example?
Material and methods:
- Line 101: the authors need to provide the sequences of all siRNA used in this study
- Line 192: what are the size of the confocal sections?
Results:
-Line 236: a quantification is expected in particular since the authors are claiming that it is over 5-fold the endogenous level.
-Line 285: regarding figures 2C and 2D, it does not seem obvious when looking at the images. Could the authors show the actual quantification plots (as described in their methods) and better explain in their figure legend which is what?
-Line 291: same comments with experiments made on MCF10a cells
-Line 296: it would be more than appreciated to see the flow results
- Line 343: What would happen if the authors were using an inducible promoter to overexpress CD82? This way they could monitor their data relative to the level of CD82 overexpression? This is a general comment which could be applied to most of their data.
-Line 348: We would like to see comparisons of cytochalasin D treatment between HB2 and HB2-CD82 cells in order to see if all CD82 function is happening through actin polymerization
- Line 359: How the authors can be sure that the differences observed are not due to differences based on the positioning of the section analyzed relative to the whole cell? The method of quantification is not described. Are they counting through several sections? Some 3D reconstruction and/or experiments like tomography could help to reinforce their data.
-Line 374: TIRF analyses could also be one way to answer my previous question, but some quantifications of TIRF analyses are then required.
- Line 388: again some quantifications would be nice here
- Line 430: how is 75% of silencing obtained? Detail how you measure it
- Line 457: Figure 5B and , Could the authors show more convincing pictures because as it stands it seems to go the over directions as the one shown after quantification
- Line 477: the entire analysis on the expression of targets of YAP is quite confusing: Why are there no changes of expression regarding Cyr61 and CTGF in HB2 cells as compared to HB2 CD82 whereas those 2 genes are strongly downregulated in MCF10a cells upon CD82 knockdown by siRNA? Also, AREG seems to be the least impacted by CD82 knockdown in MCFS10a cells whereas it is the most strongly upregulated in HB2 CD82 cells?
- In order to demonstrate in a more convincing way the links between CD82 and YAP localization and activity, could the authors correlate the changes of gene expression with changes in YAP localization?
-Line 491: regarding this last experiment, in MCF10a cells, the direct comparison with siCD82 is missing and in the HB2 system we are missing the comparison with HB2 cells to really see if the talin area is back to the level observed without CD82 expression. Measurement of speed and directionality would be nice, alongside movies regarding YAP experiments.
Discussion:
-Line 534, the function of other tetraspanins such as TSPAN7 should be mentioned, in particular regarding the described relation with membrane tension, cortical actin and actin nucleation.
-Line 567: The data as of now are not convincing enough to make such a claim. Some links are observed, but no direct demonstration. Could the authors comment on how a tetraspanin could influence the dynamism of caveolae and why then it should be specific to CD82 and not a function of the tetraspanin web?
Minor comments:
Introduction
- Affiliation numbers are missing for most of the names
- Line 33: “mandatory” may be replaced by “necessary”
- Line 65 “led us speculate” should be “led us to speculate”
-Line 71: in their introduction, the authors are not mentioning at all the role(s) played by actin nucleation (branching of actin filaments), in regards to membrane tension and in particular the role played by cortical actin. It would be great to comment on this aspect of actin.
Results:
- Line 256: in figure S2E, could the authors comment on the results regarding the impact on speed? There seems to be a trend, but maybe not enough measurements made to get statistically significant results?
- Line 260: The legend is missing for panel D of Figure 2.
- Line 309: and what about actin nucleation? Authors are not showing any data on the impact of actin nucleation (ARP2/3)
- Line 304: the impact on actin polymerization specifically was not really shown. Maybe rephrase differently.
-Line 430: it is either Figure 5A but then all other main figures need to be changed or figure 5 should be renamed as S5 line 426
Discussion:
-Line 513: the legend describing the right part is missing. Did the authors make this figure? Why is the figure legend embedded in the figure screenshot?
Reviewer 2 Report
cells-1192922
Mechanical control of cell migration by the metastasis suppressor tetraspanin CD82/KAI1.
In this manuscript, Ordas et al, have presented nice evidences of tetraspannin CD82 in HB2 cells. They have constructed this HB2 and HB2-CD82 cell pair, and also used MCF10a cells. In HB2 cells, CD82 apparently regulates persistent migration of cells. Stress fibers and focal adhesion sizes and dynamics are shown to be different in HB2 and HB2-82 cells. Membrane tension is regulated by CD82, which was confirmed in MCF10a cells. Mechanosensing YAP appeared to be affected by CD82, which then results in changes in membrane tension, cell surface caveolae abundance YAP translocation. The authors concluded that CD82 controls 2D cell migration using membrane-driven mechanics involving caveolin and the YAP pathway.
This is a reasonably clearly presented manuscript and is publishable upon minor revisions.
- I am not very familiar with the data analysis method used for example in Fig. 1C, Fig. 3B where statistical significance is shown. If someone know better can help to check that, it will be more assuring.
- Some discussion of extrapolation of the results from those cell lines to other cell types would be helpful.
- I did not find the suppl videos, but I assume that they are fine.
Minors.
I found some scale bars and labels are just too small to see.
Fig. 6 legend. “Lift” is used, then using “Right” would help clarity.
Ref. 27 is not shown in correct format.
Reviewer 3 Report
This paper provides interesting information on the role of tetraspanin CD82 (a metastasis suppressor) in cell migration through its regulation of membrane tension, cell surface caveolae and YAP nuclear translocation. The paper is generally well written, with the basis of the work clearly described and the results well-presented and discussed. It represents an important contribution to our understanding and is worthy of publication. There are, however, some minor errors that need to be addressed.
- There are a number of mistakes in English (spelling, punctuation, misuse of capitals and tenses) throughout and references are not always cited correctly. The manuscript therefore needs some careful editing.
- There are some omissions in the Methods section:
- The medium the cells are grown in isn’t stated and the source of the MCF10a cell line isn’t given.
- The source of the antibodies used for Western blotting doesn’t appear to be given
- In Figure 2A and C, the legend doesn’t correspond to the position of the labelled photographs. The key from 2D is missing (but presumably corresponds to that given in Fig S3E).
- In Figure 4A, it’s quite difficult to see bars representing the control – the bars should have an outline/more distinct shade.
- Figure 6 (and the legend) is unclear. What does the right side of the figure represent? What are the inserts showing?
Round 2
Reviewer 1 Report
Most of the major concerns reported in the first review are still left unanswered for "lack of time" as mentioned by the authors. Other simple questions such as protein quantification or clarification of data could have been better answered
Round 3
Reviewer 1 Report
I would like to thank the authors for having performed quantifications of their western blot and of their analyses using miscroscopy. they have also clarified additional points regarding methods.